# New Understanding of Diagnosis, Treatment and Prevention of Endometriosis

**DOI:** 10.3390/ijerph19116725

**Published:** 2022-05-31

**Authors:** Bedayah Amro, Maria Eugenia Ramirez Aristondo, Shaima Alsuwaidi, Basma Almaamari, Zeinab Hakim, Muna Tahlak, Arnaud Wattiez, Philippe R. Koninckx

**Affiliations:** 1Latifa Hospital, Dubai 9115, United Arab Emirates; bedaya1984@yahoo.com (B.A.); mearistondo@gmail.com (M.E.R.A.); shaima_alsuwaidi1@hotmail.com (S.A.); dr.b.almaamari@gmail.com (B.A.); zeinabhakim@icloud.com (Z.H.); munatahlak@gmail.com (M.T.); arnaud.wattiez@wanadoo.fr (A.W.); 2Department of OBGYN, Faculty of Medicine, University Strasbourg, 6081 Strasbourg, France; 3Department of OBGYN, Faculty of Medicine, Katholieke University Leuven, 3000 Leuven, Belgium; 4Department of OBGYN, Faculty of Medicine, University of Oxford, Oxford OX1 2JD, UK; 5Department of OBGYN, Faculty of Medicine, University of Cattolica, 20123 Milano, Italy; 6Department of OBGYN, Faculty of Medicine, Moscow State University, 119991 Moscow, Russia

**Keywords:** endeometriosis, adenomyoisis, genetic and epgigenetic, surgery, medical therapy

## Abstract

For 100 years, pelvic endometriosis has been considered to originate from the implantation of endometrial cells following retrograde menstruation or metaplasia. Since some observations, such as the clonal aspect, the biochemical variability of lesions and endometriosis in women without endometrium, the genetic-epigenetic (G-E) theory describes that endometriosis only begins after a series of cumulative G-E cellular changes. This explains that the endometriotic may originate from any pluripotent cell apart from the endometrium, that ‘endometrium-like cells’ can harbour important G-E differences, and that the risk is higher in predisposed women with more inherited incidents. A consequence is a high risk after puberty which decreases progressively thereafter. Considering a 10-year delay between initiation and performing a laparoscopy, this was observed in the United Arab Emirates, Belgium, France and USA. The subsequent growth varies with the G-E changes and the environment but is self-limiting probably because of the immunologic reaction and fibrosis. That each lesion has a different set of G-E incidents explains the variability of pain and the response to hormonal treatment. New lesions may develop, but recurrences after surgical excision are rare. The fibrosis around endometriosis belongs to the body and does not need to be removed. This suggests conservative excision or minimal bowel without safety margins and superficial treatment of ovarian endometriosis. This G-E concept also suggests prevention by decreasing oxidative stress from retrograde menstruation or the peritoneal microbiome. This suggests the prevention of vaginal infections and changes in the gastrointestinal microbiota through food intake and exercise. In conclusion, a higher risk of initiating endometriosis during adolescence was observed in UAE, France, Belgium and USA. This new understanding and the limited growth opens perspectives for earlier diagnosis and better treatment.

## 1. Introduction 

Endometriosis was described more than 100 years ago as “endometrial-like tissue” outside the uterus [1,2]. Initially, these were accidental findings during surgery, but already in 1960, endometriosis had become the main cause of surgery in women [3]. After the introduction of laparoscopy in the 1970s, it was realised that typical pelvic endometriosis was very frequent in women with infertility or pain. In 1986 [4], unstained lesions were recognised as endometriosis, and in 1990 [5], smaller deep endometriosis lesions were described. Extrapelvic endometriosis occurs in the lungs, although occasionally, endometriosis can be found in many other places of the body, such as the nose [6]. 

Macroscopically, abdominal endometriosis presents as superficial, cystic ovarian or deep endometriosis. Superficial endometriosis is 1–3 mm, unstained, histologically active subtle lesions or 1–3 cm, black ‘shotgun’, inactive, typical lesions in a white fibrotic area. Ovarian cysts, filled with ‘chocolate’ fluid, ranging from 1 to more than 10 cm in diameter and deep endometriosis lesions are solid-tissue tumours from a few mm to more than 5 cm in diameter. However, it should be realised that deep endometriosis is poorly defined; if all lesions deeper than 5 mm under the peritoneum [7] are considered deep endometriosis, many larger typical lesions [8] risk being classified as deep endometriosis. 

Clinically, superficial pelvic endometriosis is associated with mild pain, cystic ovarian endometriosis with severe and deep endometriosis with very severe pain. However, some 50%, 25% and 5% of these women are pain-free, respectively. Endometriosis is associated with infertility, but it is unclear whether endometriosis causes infertility, except in cystic ovarian endometriosis with severe adhesions. 

Over the last decades, we progressively understood that cells with similar histological aspects could hide important genetic, epigenetic and molecular biological differences. Furthermore, our understanding of the pathophysiology of endometriosis has changed significantly, and the impact on diagnosis and treatment of endometriosis will be reviewed.

## 2. New Understanding of Pathophysiology 

According to the 100-year-old Sampson theory [1], endometriosis was considered a consequence of retrograde menstruation and implantation of endometrial cells. However, this hypothesis is no longer acceptable as the main cause of endometriosis. Implantation of the endometrium is not compatible with the observation that each endometriosis lesion is clonal, i.e., grown from one progenitor cell [8], and that a woman with 10 lesions has 10 different clones. It thus is not surprising that individual lesions can be different, harbouring variable cancer driver mutations and biochemical differences such as a variable progesterone resistance or aromatase activity [9]. The implantation theory has difficulty explaining the different types of endometriosis lesions, the hereditary aspect [10] and endometriosis initiating in postmenopausal women [11] (Figure 1), in women without a uterus [12] or in men [13]. Moreover, since almost all women have retrograde menstruation, implantation and endometriosis would be expected in most, if not all, women [14]. 

Although still widely accepted, the limitations of the implantation theory were challenged from the beginning by the metaplasia theory. Metaplasia started as an old histological concept describing that the histological aspect of a mature cell could change into that of another mature cell. Much later, this was broadened to the development of stem cells. In contrast with changes in DNA that are irreversible, the reversibility of epigenetic changes is much less clear. Our cells differentiate by hiding, using or amplifying specific stretches of DNA. Epigenetics, defined as non-DNA transmissible changes, can be compared with origami, folding a sheet of paper into a figure. After unfolding, the paper is still intact, but to fold the same figure again has become easier, explaining the transmission of epigenetic information after cell division. Epigenetic changes thus are theoretically reversible, such as an origami folded figure, but reversibility decreases with the complexity of the changes because of the increased risk of mistakes when unfolding. The difference between reversible and irreversible epigenetic changes thus is no longer absolute but rather an estimation of risk. Not yet understood is how epigenetics are transmitted transgenerationally, as observed for the predisposition to music. 

The recently proposed G-E theory [8] postulates that a series of cumulative G-E incidents are needed before endometriosis starts to develop. Each cell division carries a risk of mistakes, and the risk probably increases with pollution such as dioxin, ionising radiation or oxidative stress. The endometrium, being the fastest growing tissue of our body, therefore has an increased risk. Furthermore, specific for endometriosis is the oxidative stress of the peritoneal cavity because of retrograde menstruation and microbiota [15,16]. Since retrograde menstruation causes an inflammatory reaction because of blood and iron, it is not surprising that more abundant retrograde menstruation has been linked to endometriosis. It is still unclear how the peritoneal microbiome causes endometriosis, but women with endometriosis have a different peritoneal microbiome, more vaginal infections, more pelvic inflammatory disease and endometriosis lesions harbour unexpected evidence of HPV and Shigella infections. Eventual incidents add to the incidents inherited at birth, explaining predisposition and heredity. Examples of DNA mutations are the frequent cancer driver mutations in endometriosis [17,18,19] secondary to genomic instability. Specific epigenetic incidents are not yet documented. A difficulty is that epigenetic reorganisations can be too complex or unlikely to be reversible, although nothing is broken. Therefore, the most important in the G-E theory is that the endometriotic cell has undergone irreversible G and/or E changes making endometriosis permanently different from the endometrium, notwithstanding an eventual similar histological aspect. More speculative today is that these endometriotic cells could induce in the surrounding cells -similar to cancer- reversible metaplasia making them look histologically such as endometrium. Although not formally proven, this is strongly suggested when considering that recurrences do not increase following resections without safety margins.

The G-E theory explains that endometriosis can arise from any poorly differentiated cell, such as stem cells or bone marrow cells. However, the likelihood of developing from the endometrium or embryological rests, already developed in that direction, is higher. 

The redundancy of molecular biological pathways explains that not every new incident is visible. A problem with one of the brake circuits of a car that has two is not obvious. Only after the accumulation of several incidents, over-ruling the capacity of the cell to cope, endometriosis lesions develop. This explains that the number and type of G-E incidents are different for each endometriosis lesion, being clonal with differences in their molecular biology, explaining the variable degree of aromatase activity or progesterone resistance [9,20]. This variability also explains that not all lesions are painful or cause pain at a distance [21] and that the response to hormonal treatment can be different [22,23]. The specific combination of G-E incidents can also explain why lesions develop as superficial, cystic or deep endometriosis. Furthermore, important is that the many endometriosis-associated changes in endometrium, infertility, immunology and pregnancy are not necessarily a consequence of endometriosis but can be explained by the inherited predisposition. Heredity thus could be the common cause of both endometriosis and these changes, which therefore are associated. That increased CA125 concentrations return to normal after endometriosis excision suggests these are a consequence of endometriosis. That decreased NK activity [24] and increased pregnancy complications [25] do not change after surgery, suggesting they are pre-existing to endometriosis. The G-E pathogenesis also explains that deep endometriosis can occasionally initiate and develop many years after menopause in women not taking estrogens (Figure 1) [11]. 

After initiation, the growth of endometriosis lesions varies with the specific combination of G-E incidents and the environment of the peritoneal cavity [26], which is a specific microenvironment with estrogens, progestins, growth factors, cytokines and the immune response [7] different from plasma. The peritoneal cavity in endometriosis is comparable with low-grade inflammation, with increased numbers of activated macrophages and their secretion products. Growth is moreover associated with immunology and inflammation-driven fibrosis that can decrease vascularisation and stop the growth [8] (Figure 2). The result can be compared to a war of trenches: the army cannot get in, but the enemy cannot get out. This self-limiting growth explains that the severity of endometriosis lesions does not increase with age [27]. It remains unclear what part of the immunologic reaction is a consequence of endometriosis and what part is caused by the genetic predisposition to endometriosis. Self-limiting growth of endometriosis is consistent with the clinical observations that most deep endometriosis nodules are not operated on whenthey cause little pain and do not seem to grow over time (Jörg Keckstein and PK, personal communication). 

The peritoneal cavity is not sterile but contains a microbiome originating from the vagina, uterus and fallopian tubes and from the gut by transmural migration. Microbiota and associated factors seem important for the G-E origin and the growth of endometriosis [6]. This could explain the effect of food intake [28,29,30] and exercise [31,32,33,34,35,36,37] on endometriosis and of food supplements such as berberine [38,39], NAC [40], zinc [36,37] and gluten-free diet [41]. 

Finally, the observations on progesterone resistance in the basalis of the endometrium [42] together with the high steroid hormone concentrations in the peritoneal cavity [43] after ovulation are not yet fully understood. However, since superficial endometriosis is mainly influenced by peritoneal fluid concentrations, the mechanism of medical therapy, suppressing ovulation and thus decreasing estrogen and progesterone concentrations, needs to be reconsidered [9].

## 3. Clinical Consequences of Our New Understanding

### 3.1. Higher Risk of Initiating Endometriosis during Adolescence

A logical consequence of the G-E theory of endometriosis is that the risk of initiating endometriosis increases after puberty because of the hormonal changes, the oxidative stress of retrograde menstruation and the changed peritoneal microbiota. Susceptible women with more inherited mistakes risk initiating endometriosis earlier, and the remaining groups will have a progressively lower risk (Figure 3). Endometriosis needs to grow for a few years before becoming symptomatic, which is expressed by the well known diagnostic delay of up to 10 years before a laparoscopy is performed [44,45]. The risk of initiating endometriosis after puberty and a 10-year delay is compatible with the observation that most laparoscopies for endometriosis are performed in women between 25 and 30 years of age. That the remaining group will have a progressively lower risk is compatible with an exponential decline thereafter. This is comparable to fertility: the most fertile couples have a high probability of conception, and the remaining group will progressively become less fertile. 

This concept of a higher risk during adolescence with a progressive decline thereafter is important for understanding the natural history of endometriosis. We, therefore, evaluated the incidence of laparoscopies for endometriosis at different ages in several populations. All laparoscopies for endometriosis performed at Latifa Hospital, in Dubai, the United Arab Emirates, during 2020–2021 (n = 132) and their severity were retrieved. These data confirm the observation that most laparoscopies were performed between 25 and 35 years of age, with a progressive decline thereafter. This is comparable to the observations in France [46], Belgium [27] and the USA [47]. The increased risk of initiating endometriosis during adolescence thus seems similar in Arabic and western countries, notwithstanding differences in climate, food intake and social organisations. However, the severity of endometriosis in the different populations was slightly different, with more severe endometriosis in Latifa Dubai. Since this might reflect a referral bias, the periods 1990–2000 and 2001–2011 (multivariate logistic regression, Proc logistic, SAS) in Belgium were compared, demonstrating that in the latter group, laparoscopies occurred some 5 years later (*p* < 0.0001), and that the lesions were more severe (*p* = 0.01) (Figure 4) similar to the observations in Dubai. 

### 3.2. Clinical Symptoms of Endometriosis

Endometriosis causes variable pain (Figure 5). Superficial lesions are associated with moderate pain in 50% of women. This can be understood since, by conscious pain mapping, only 30% of subtle and typical lesions are painful. Clinically more important is that these lesions cause pain in the surrounding peritoneum or neuroinflammation up to 3 cm distance [10,11]. Thus, considering distances in the pelvis, sympathetic and most larger nerves are within a 3 cm distance, explaining that cyclic sciatalgia can be cured by surgical treatment of peritoneal pockets [21]. Cystic ovarian endometriosis causes severe pain in 70–80% of women. Chocolate cysts that are not painful, especially those without adhesions and with an acute onset, rather suggest cystic corpora lutea. Deep endometriosis is generally very painful, but an estimated 5% is not. Important is that there is little relationship between the size of the lesions and the severity of the pain. This could be explained by G-E differences and by the poorly understood innervation of deep endometriosis. Because of this variability of pain in ‘endometrium like lesions’, statisticians recently suggested redefining endometriosis as ‘symptomatic endometrium like lesions” [48], which is fully compatible with variable epigenetic pathophysiology. 

The endometriotic pain is localised in the hypogastric area and iliac fossa, with frequent radiation to the lower back. Radiation to the anterior and inner surface of the thigh indicates ovarian involvement, while perineal radiation is a sign of colon involvement in the last 20 cm ([49] and personal communication PK). The type of pain varies from chronic pain, dysmenorrhea, dyschezia and deep dyspareunia in low deep endometriosis.

It is unclear whether endometriosis causes infertility except when adhesions in cystic ovarian endometriosis involve the oviducts. That women with more than 1 year of infertility have typical and subtle lesions in more than 50%, rather suggests that the inherited G-E defects cause a predisposition to develop endometriosis and also decrease fertility. Endometriosis and infertility can thus be associated because of common hereditary causes. The mechanisms of the associated infertility are unknown, and speculation ranges from changes in the endometrium to changes in the reproductive tract and peritoneal microbiota. A common hereditary factor also explains that endometriosis-associated preeclampsia and small-for-dates babies do not improve after deep endometriosis excision [50]. The mechanism of the association of endometriosis with a wide range of other symptoms, such as depression, lower BMI, autoimmune diseases, allergies, etc., are not clear. However, considering the recently observed associations between the intestinal microbiome and a whole range of effects on the brain, immunology, and rheumatoid arthritis it is attractive to speculate that many of these associations with endometriosis might also find their origin in hereditary factors increasing the risk of endometriosis and changing the bowel microbiota as reviewed in 42 articles in 2021 only [51,52,53,54,55,56,57,58,59,60]). The association with sexual dysfunction seems to be caused by the pain, which is a strong inhibitor of sexuality [61,62,63,64]. 

### 3.3. Diagnosis

The diagnosis of superficial pelvic endometriosis requires laparoscopy. The diagnosis of cystic ovarian endometriosis can be made with ultrasonographic or MRI imaging. Large and low deep endometriosis lesions can be felt clinically [65], but most deep endometriosis lesions require a laparoscopy for diagnosis. The value of imaging in diagnosing deep endometriosis is controversial and not well documented for smaller lesions. Biomarkers are not useful.

The controversy of diagnosing deep endometriosis by imaging results from the interpretation of the accuracy of diagnostic tests, which varies when prevalences of the disease are taken into account. Positive (PPV) or negative predictive values (NPV) quickly drop when prevalences are low, especially below 5%. Sensitivity and specificity of more than 95% might seem a good test, as observed for deep endometriosis, but with a prevalence of 5%, the predictive values are only 60% [14]. Another example is that a test with 99% sensitivity and 99% specificity for a disease occurring in 1% results in as many false positives as true positives. For the same reason, the predictive values of imaging are higher in referral centres with a higher incidence. Unfortunately, the accuracy for smaller nodules less than 1 cm is not known.

Imaging, therefore, should be used carefully as an indication for laparoscopy unless lesions of deep endometriosis are large [15,16]. However, imaging is important for preoperative assessment and informed consent and counselling as it provides an idea of the extent and the surgical difficulty of the larger lesion. It is not clear whether imaging changes the quality or type of surgery, although widely believed.

The indication for laparoscopy varies widely in the literature. We consider that the indication should be a clinical judgement based on the severity and localization of the pain. This is consistent with the observation by Bayesian statistical analysis [17] that the added value of imaging for the indication or type of intervention is marginal.

Another problem with imaging of deep endometriosis is the variable description of the size of lesions. In order o calculate a volume, three dimensions are needed. However, many imaging reports list only two diameters. The confusion in the literature is highlighted as follows: a 3 × 3 × 3 cm sphere has a volume of 14 mL, which is much larger than 3.5 mL for a 3 × 3 × 0.5 cm cylinder.

### 3.4. Treatment 

The new understanding of the pathophysiology of endometriosis will change medical therapy with oestro-progestins, progestins or GN-RH agonists or antagonists. The variability of lesions explains that the pain relief is minimal or absent in 10% to 20% [23,66], although effective in some 70% of women. Therefore, it is not reasonable to continue the same treatment if the effect is unsatisfactory after a few months. The G-E variability of lesions also explains that lesions can continue to grow during medical treatment, as observed after menopause [11]. Therefore it has been suggested to evaluate treatment by ultrasound every 6 months [67]. More fundamentally, if lesions are heterogeneous, traditional statistical analyses are inappropriate to evaluate the results of medical treatment since traditional statistics require a homogeneous population. 

Also, the mechanism of action of medical treatment is unclear. Medical treatment abolishing ovulation decreases the estrogen and progesterone concentrations, especially in the peritoneal cavity, where they are normally much higher than in plasma. Considering the high concentrations of progesterone in peritoneal fluid, it is questionable whether progesterone has a specific effect on superficial endometriosis [9]. It, therefore, is also questionable whether some progestins are more effective than others. 

Considering endometriosis, a disease with G-E changes, possibly inducing endometrium such as metaplasia in the surrounding cells, and eventually at a distance as demonstrated for inducing pain in the surrounding peritoneum up to 3 cm, is important when considering surgical excision. This induced reversible metaplasia can explain that smaller endometrium such as nests of cells in the bowel at a distance from a deep nodule or in lymph nodes do not cause recurrences or clinical symptoms, as suggested by short bowel resections without safety margins. This is also supported by the observation that recurrence rates are not higher following conservative excision, being likely incomplete at the cellular level, than a bowel resection. Another direct consequence of the G-E theory is that lesions are biochemically variable and that some lesions can develop with minimal estrogen concentrations, as suggested by the occurrence of endometriosis after menopause or after hysterectomy with removal of ovaries. Unfortunately, we cannot distinguish today between endometriotic and reversible metaplastic endometrium such as lesions. Therefore, the aim of surgery remains to remove all visible endometriosis lesions with minimal damage. For superficial lesions, CO_2_ laser vaporisation is still theoretically superior because of the minimal tissue damage and, thus, probably postoperative adhesions. The limited growth because of fibrosis and the immune reaction and inflammation also could change surgery [19] since the fibrosis belongs to the body and does not need to be removed, similar to an abscess that is drained without removing the surrounding fibrosis. For cystic ovarian endometriosis, capsule excision produces fewer recurrences but more ovarian damage than superficial destruction. However, the latter seems theoretically preferable because the endometriosis invasion into the fibrotic capsule is only 1–2 mm. Therefore we recommend today superficial destruction in small lesions while chemical destruction as alcoholisation deserves investigation, especially for larger lesions. 

The understanding of fibrosis also could affect the excision of deep endometriosis infiltrating the bowel or of causing ureter obstruction. Although for the bowel a conservative excision and a bowel resection remain debated, the concept of fibrosis belonging to the body suggests a more conservative approach [68], eventually leaving a rim of fibrosis or finishing an incomplete excision with a wedge resection using a circular stapler [18]. Without discussing the many aspects of bowel resection, minimal instead of large bowel resection are sufficient. For the sigmoid, the liberal use of short bowel resections (Figure 6) seems preferable over excisions since this resection has few complications, while the mobility of the sigmoid makes excision technically difficult Although for deep bladder endometriosis, the same concepts are valid, wide excision is still preferred since the bladder heals well. Furthermore, for the vaginal cuff, we prefer a wide excision, with whatever technique, since in our experience, almost all recurrences were observed in the vaginal cuff. Considering that fibrosis can cause ureteral stenosis, it seems wise to remove all fibrosis by a resection anastomosis rather than the lengthy and difficult excision around severe stenosis. A discussion of bowel preparation is beyond the scope of this manuscript, as is the eventual systematic appendectomy during deep endometriosis surgery. Not yet understood is the role of neural invasion of deep endometriosis and pelvic pain and how this affects surgical management of the surrounding fibrosis. 

### 3.5. Prevention of Recurrences

All women deserve the prevention of recurrences after surgery. Although still unproven, it is logical to assume that decreasing oxidative stress by preventing retrograde menstruation or changing the peritoneal and upper genital tract microbiome will decrease the risk of initiating endometriosis. Both can be achieved with continuous oral contraception or progestogens, decreasing menstruation and the risk of ascending infections. Moreover, vaginal infections merit more attention.

Preliminary evidence suggests that food rich in antioxidant as omega3, Vit E, Vit C and citrus decrease the risk of endometriosis [69,70]. It is too early to fully understand the effect of vitamins on inflammation and immune response in endometriosis [71] and to prevent endometriosis initiation or growth through modifications of the intestinal microbiota with food intake and sports.

## 4. Discussion of Changing Endometriosis Strategies

Adolescent endometriosis should receive more attention. The higher risk of initiating endometriosis after puberty was indirectly confirmed by the incidence of laparoscopies for endometriosis in the UAE, France, Belgium and the USA. These similar observations in Arabic countries, Europe and the USA, suggest a fundamental mechanism involving estrogens and the peritoneal microbiome, less affected by food intake or climate or environment. Although not yet investigated or proven, repetitive vaginal infections and severe dysmenorrhoea in women with a hereditary risk seem to deserve more clinical interest [21]. If, in addition, prevention could be proven to be effective, an earlier laparoscopy should be considered to prevent more severe lesions and infertility [72].

Medical therapy of endometriosis needs to be reconsidered since endometriosis lesions are heterogeneous, with variable progesterone resistance and since superficial endometriosis is affected mainly by peritoneal fluid concentrations of steroid hormones. More specifically, it needs to be investigated whether and in which lesions a specific progesterone effect can be expected.

It remains debated whether laparoscopy is useful in women with infertility and minimal or no complaints. We believe that a focused and complete workup followed by a treatment plan taking all factors into account is preferable, although some like a more sequential approach. A full discussion of the cumulative pregnancy rates following a fast versus a sequential investigation and when IVF should be considered remains debated. That the severity of endometriosis and adhesions can be quite unpredictable with eventually difficult surgery explains that an earlier laparoscopy is mainly supported by surgery-oriented groups, in contrast to IVF centres with limited surgical interest.

Women with pain but without major pathologies such as a large palpable nodule or an ovarian endometriosis cyst require a clinical decision based on their symptoms. If the only complaint is dysmenorrhea, a short trial with medical therapy to exclude spastic dysmenorrhea seems logical. However, if the improvement is insufficient, laparoscopy or transvaginal hydro laparoscopy (Figure 7) should be considered.

Once the decision to perform a laparoscopy has been made, the informed consent is based on the clinical symptoms, exam and imaging. If deep endometriosis is suspected, it is desirable to exclude hydronephrosis, eventually with an ultrasound of the kidneys and the ureters. How to exclude bowel stenosis as an indication for the type of surgery to be performed remains discussed. Colon stenosis can be assessed with a contrast enema, but the expertise has decreased over time. Virtual colonoscopy, colonoscopy and ultrasound seem less performant. Deciding to do bowel resection before surgery based on imaging and the depth of infiltration is associated with a higher percentage of bowel resections than deciding during the laparoscopy. However, the latter is impractical unless the gynaecologist/pelvic surgeon can perform the bowel resections. The same principle applies to the wedge resection of the bowel in addition to a conservative excision with a circular stapler, which is gradually becoming more important.

## 5. Conclusions

The management of endometriosis changes if considered initiating following a series of cumulative genetic-epigenetic incidents, with a subsequent self-limiting growth and fibrosis. The risk of initiating endometriosis will be highest after menarche, mainly in predisposed women, and will decrease thereafter, becoming low after 3O years. Medical therapy is suggested to prevent new lesions or recurrences; as a therapy of pain it needs to be revised because of the individual variability of lesions. Also surgery for endometriosis is bound to change if the fibrosis is not part of the disease and does not need to be removed. 

## Figures and Tables

**Figure 1 ijerph-19-06725-f001:**
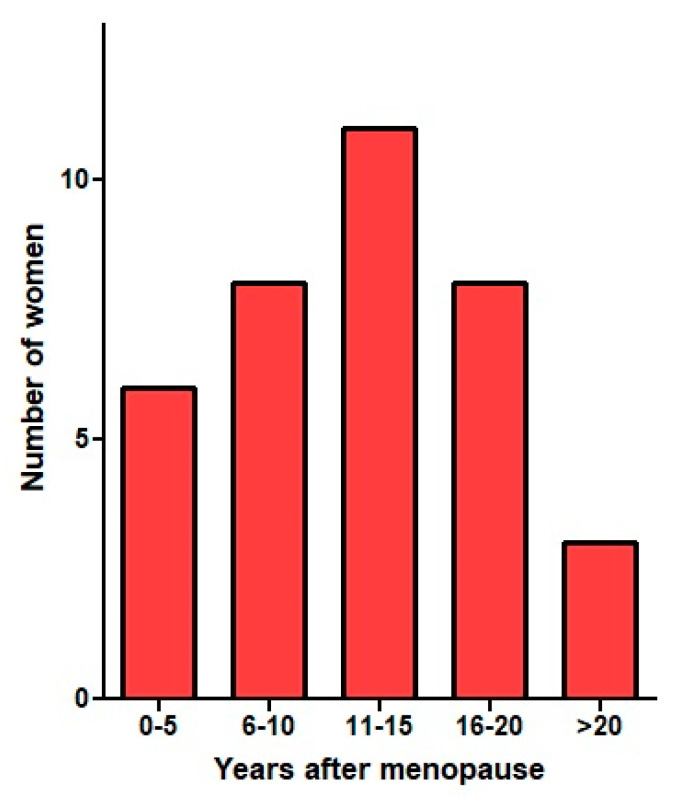
Deep endometriosis initiating more than 10 years after menopause in women not taking estrogens (reprinted from [11] with permission) suggests monitoring the growth of endometriosis during medical treatment.

**Figure 2 ijerph-19-06725-f002:**
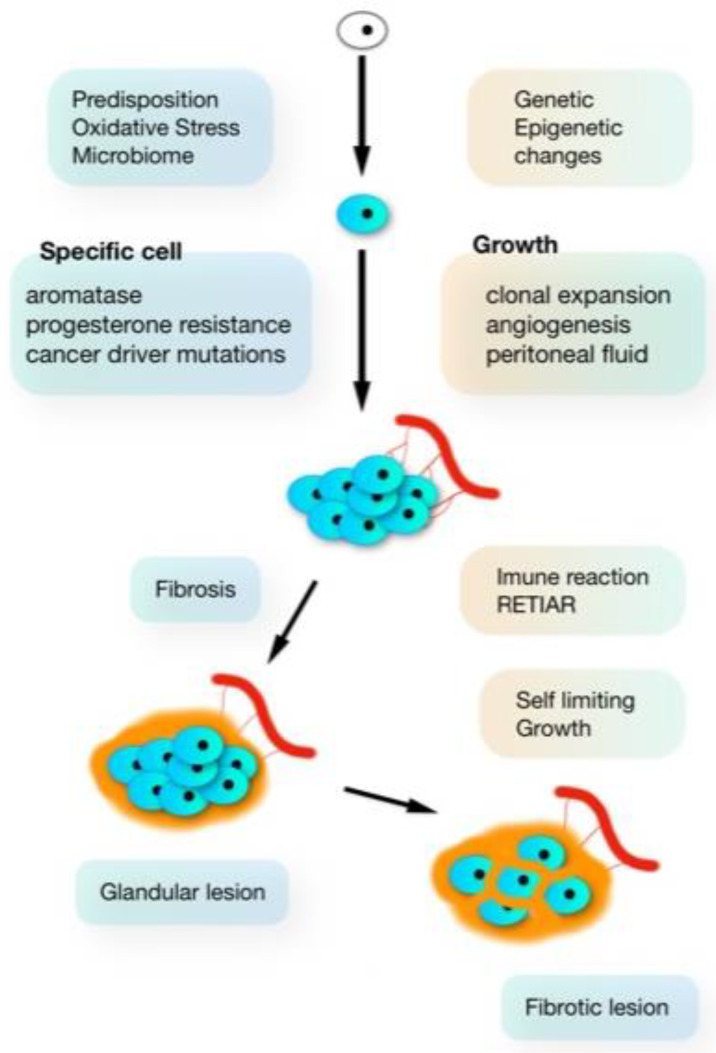
Origin and development of endometriosis: a combination of G-E events, with further growth, immune response, inflammation and fibrosis (figure from [7]).

**Figure 3 ijerph-19-06725-f003:**
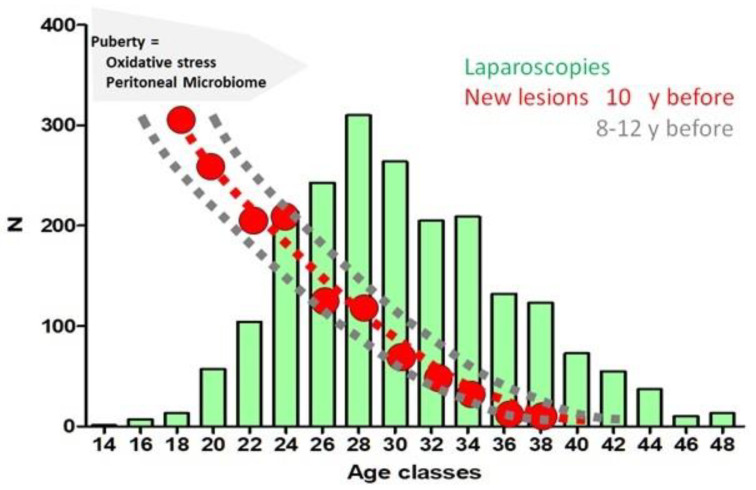
During adolescence the risk of initiating endometriosis is highest with a progressive decline thereafter. After puberty the endocrinology, and the oxidative stress of retrograde menstruation and the peritoneal microbiome change. Susceptible women will initiate endometriosis earlier while the remaining group will have a lower risk. Considering a 5 to 10 years delay between the initiation of endometriosis and the laparoscopy explains that most laparoscopies for endometriosis were performed between 25 and 30 years of age with an exponential decline thereafter (reproduced with permission [8]).

**Figure 4 ijerph-19-06725-f004:**
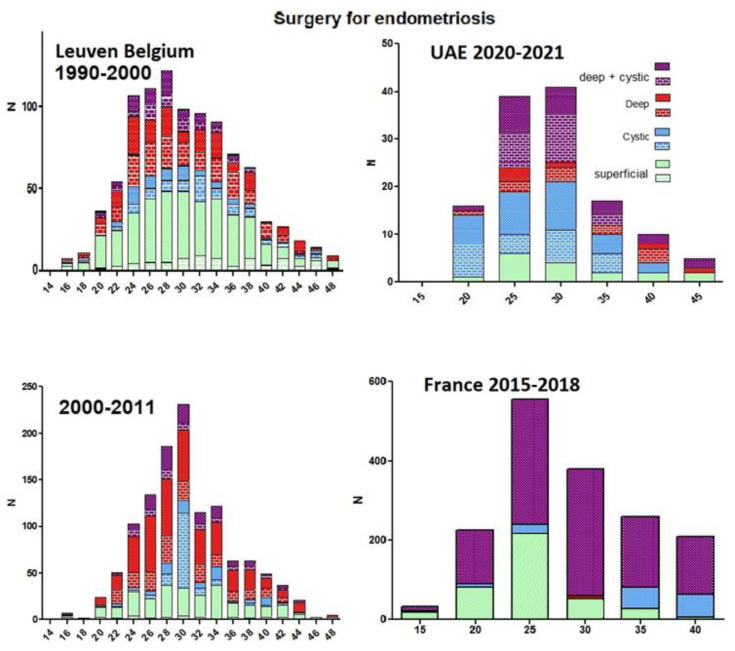
Incidence of laparoscopies and type of endometriosis In Latifa, United Arab Emirates, in France and Belgium. The severity of deep (>2 cm) and cystic (>3 cm) endometriosis is indicated by darker colours.

**Figure 5 ijerph-19-06725-f005:**
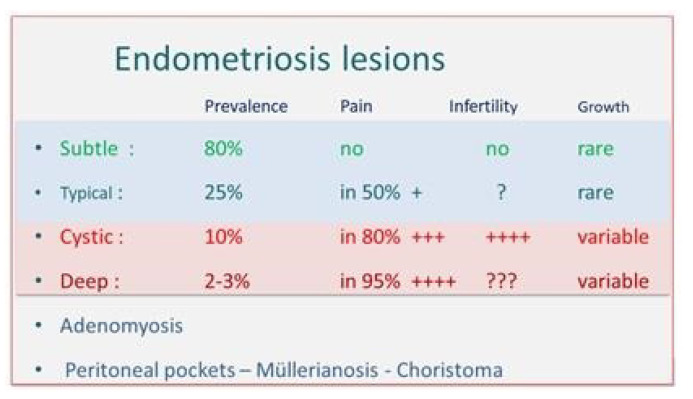
Endometriotic lesions as a cause of pain and infertility. Adenomyosis, peritoneal pockets and Müllerianosis are not discussed.

**Figure 6 ijerph-19-06725-f006:**
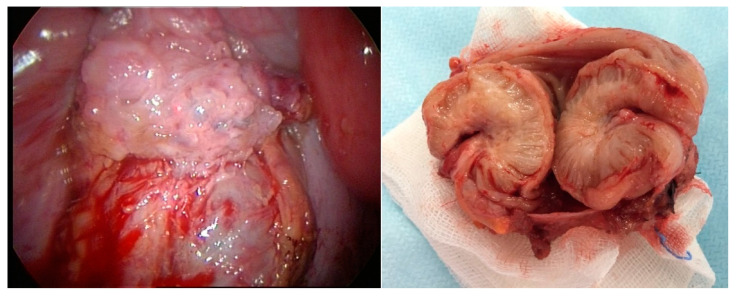
Images of conservative excision of short bowel resection in deep endometriosis in the bowel. Note the fibrosis around the endometriosis.

**Figure 7 ijerph-19-06725-f007:**
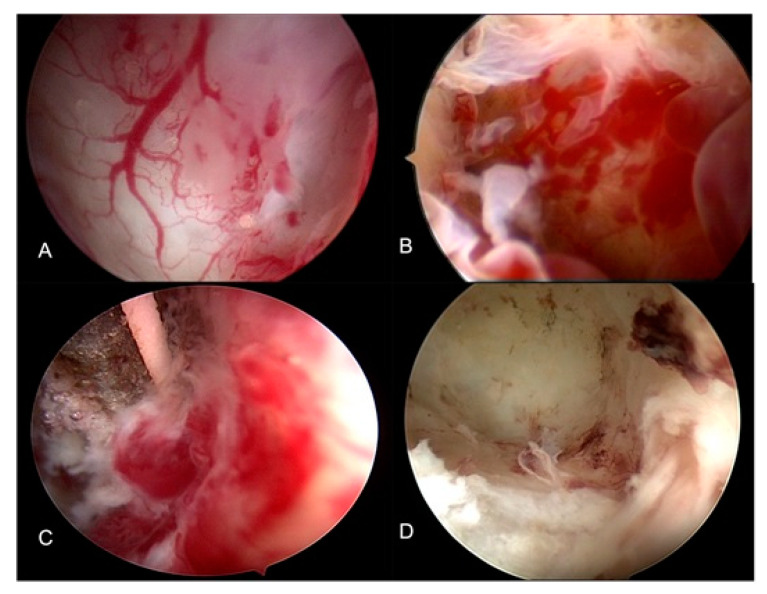
Transvaginal hydro laparoscopy images of an endometrioma before (**A**) and after (**B**) surgery of endometriosis. (**A**,**B**) ovarioscopic view: note neoangiogenesis and endometrium-like tissue. (**C**) superficial coagulation using 5Fr bipolar probe. (**D**) Final result after coagulation: minimal trauma, no carbonization [67].

## Data Availability

All data are available as anonymous SAS files from the corresponding author.

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
