# Peer review of "New Understanding of Diagnosis, Treatment and Prevention of Endometriosis"

_ijerph, 2022, doi:10.3390/ijerph19116725_

Round 1

Reviewer 1 Report

Thank you very much for the invitation to review of the manuscript. It a great pleasure for me.

The purpose of study of Amro et al. was to presence new view of diagnosis, treatment and prevention of  endometriosis.

The topic is very interesting, however I have a few questions and suggestions:

  1. The abstract is too long, some information are not important enough to put them to the abstract section.
  2. The first sentence in the Introduction section, the Authors start from “adenomyosis”. Then they don't mention it anymore. Maybe better not to mention it or describe the differences between these diseases?
  3. The Authors focus on the genetic-epigenetic theory of the development of endometriosis. Maybe it is worth explaining the theory better? What changes may there be? what is the consequence of this?
  4. It is worth adding information about the distant occurrence of endometriosis foci, not only within the peritoneal cavity, which also refutes Sampson's theories. Likewise, the appearance of endometriosis outbreaks in women after hysterectomy.
  5. The authors do not mention this treatment for endometriosis. Hysterectomy is very popular and often offere to women with endometriosis, e.g. in the UK.
  6. Is there a correlation between changes in EG and the severity of the disease?
  7. The treatment section provides information on the therapy systems currently available. Please state how the genetic-epigenetic theory can modify these schemas. Are there any clinical studies based on this?
  8. An addition, is there any research being done on the use of biological treatments?
  9. What are the guidelines for treating pain in endometriosis?
  10. I really like the prevence chapter. However, the authors speak of "it is logical to prevent oxidative stress by preventing retrograde menstruation", although they previously stated that the retrograde menstrual flow does not have to be the cause of endometriosis? Could they explain it? How can estrogen control be related to prevention?
  11. In the case of references, self-citation is very large, especially as it is based on review papers.

Author Response

Referee 1

We do thank you for these constructive comments

  1. The abstract is too long, some information are not important enough to put them to the abstract section.

Thank you. The abstract was revised extensively and shortened

  1. The first sentence in the Introduction section, the Authors start from “adenomyosis”. Then they don't mention it anymore. Maybe better not to mention it or describe the differences between these diseases?

Thank you: adenomyosis was removed. It pointed to the historically very first descriptions by Rokitanski and to the probable similarity of both diseases.  

  1. The Authors focus on the genetic-epigenetic theory of the development of endometriosis. Maybe it is worth explaining the theory better? What changes may there be? what is the consequence of this?

L124-131 we added “A difficulty is that epigenetic reorganisations can be too complex or unlikely to be reversible, although nothing is broken.  Therefore, the most important in the G-E theory is that the endometriotic cell has undergone irreversible G and/or E changes making endometriosis permanently different from the endometrium notwithstanding an eventual similar histological aspect. More speculative today is that these endometriotic cells could induce in the surrounding cells -similar to cancer- reversible metaplasia making them look histologically like endometrium. Although not formally proven, this is strongly suggested when considering that recurrences do not increase following resections without safety margins.

The clinical consequences will be the focus of this manuscript.

  1. It is worth adding information about the distant occurrence of endometriosis foci, not only within the peritoneal cavity, which also refutes Sampson's theories. Likewise, the appearance of endometriosis outbreaks in women after hysterectomy.

Thank you for this important comment. We rephrased L289-303 as follows “Considering endometriosis a disease with G-E changes, possibly inducing endometrium like metaplasia in the surrounding cells, and eventually at a distance as demonstrated for inducing pain in the surrounding peritoneum up to 3 cm,  is important when considering surgical excision.  This induced reversible metaplasia can explain that smaller endometrium like nests of cells in the bowel at distance from a deep nodule or in lymph nodes do not cause recurrences or clinical symptoms, as suggested by short bowel resections without safety margins. This is also supported by the observation that recurrence rates are not higher following conservative excision, being likely incomplete at the cellular level, than a bowel resection. Another direct consequence of the G-E theory is that lesions are biochemically variable and that some lesions can develop with minimal estrogen concentrations as suggested by the occurrence of endometriosis after menopause or after hysterectomy with removal of ovaries.

  1. The authors do not mention this treatment for endometriosis. Hysterectomy is very popular and often offere to women with endometriosis, e.g. in the UK.

L 298-300 “Another direct consequence of the G-E theory, is that lesions are biochemically variable and that some lesions can develop with minimal estrogen concentrations as suggested by the occurrence of endometriosis after menopause or after hysterectomy with removal of ovaries

This is an important topic, especially when removing the ovaries. However, we did not speculate on this because of our poor understanding of the role of peritoneal fluid and progesterone resistance.

  1. Is there a correlation between changes in EG and the severity of the disease?

This can unfortunately not yet be answered. Personally, we expect a correlation with severity, as I think to know is suggested indirectly by  data of Linkage analysis

  1. The treatment section provides information on the therapy systems currently available. Please state how the genetic-epigenetic theory can modify these schemas. Are there any clinical studies based on this?

Thank you for this question. L279-282 was confusing. This was changed to “More fundamentally if lesions are heterogeneous, traditional statistical analyses are inappropriate to evaluate the results of medical treatment since these statistics require a homogeneous population. 

  1. An addition, is there any research being done on the use of biological treatments?

I fear I do not understand well the question. There is a vast and fast-growing literature on diet and endometriosis, diet and microbiota of the bowel and the peritoneal cavity and physical exercise. However, to the best of our knowledge, these data are not well understood yet. Also the role of immunology is not clear yet.

  1. What are the guidelines for treating pain in endometriosis?

We tried to stay away from guidelines and evidence-based medicine because of the actual debate on p-values, significances, evidence and Bayesian statistics. We are preparing a manuscript discussing this topic.  

  1. I really like the prevence chapter. However, the authors speak of "it is logical to prevent oxidative stress by preventing retrograde menstruation", although they previously stated that the retrograde menstrual flow does not have to be the cause of endometriosis? Could they explain it? How can estrogen control be related to prevention?

Thank you. Preventing menstruation prevents retrograde menstruation and oxidative stress in the peritoneal cavity and likely the peritoneal microbiome. In addition, decreasing ovarian activity will decrease strongly estrogen and progesterone concentrations in peritoneal fluid. As written recently in HR, we think that the decreased concentrations of estrogens in peritoneal fluid constitute the most important mechanism of medical treatment of endometriosis.

L337-342 have been changed into “All women deserve the prevention of recurrences after surgery. Although still unproven, it is logical to assume that decreasing oxidative stress by preventing retrograde menstruation or changing the peritoneal and upper genital tract microbiome will decrease the risk of initiating endometriosis. Both can be achieved with continuous oral contraception or progestogens decreasing menstruation and the risk of ascending infections. Besides, vaginal infections merit more attention.”

  1. In the case of references, self-citation is very large, especially as it is based on review papers.

Thank you for the comment. We removed all references that did not describe new aspects or elements (ref 4 and 8) not expressed in any of the previous papers.

Reviewer 2 Report

The rewiew does not add substantial contribution to the current literature on the topic. 

The manuscript is relevant for the field but is not clear and presented in a bad-structured manner. The manuscript is not scientifically sound and is not the experimental design appropriate to test the hypothesis.

The methods section and details given absent like as the statistical analysis from specific databases.

The data don’t properly show, not clear and confuse interpreted throughout the manuscript.

Author Response

Referee 2

The rewiew does not add substantial contribution to the current literature on the topic. 

The manuscript is relevant for the field but is not clear and presented in a bad-structured manner. The manuscript is not scientifically sound and is not the experimental design appropriate to test the hypothesis.

The methods section and details given absent like as the statistical analysis from specific databases.

The data don’t properly show, not clear and confuse interpreted throughout the manuscript.

Unfortunately, there seems to be a misunderstanding. This manuscript does not describe new research. We briefly review comprehensively a series of data published over the last few years. This manuscript details the clinical consequences for diagnosis and therapy of endometriosis, if considered to be caused by genetic-epigenetic changes.

The identification of the underlying G-E incidents and the molecular-biochemical mechanisms and the resulting immunologic changes will take many years to come. However, since the implantation of endometrial cells has become unlikely to be the main cause of endometriosis, it seems wise to adapt our management accordingly.

Reviewer 3 Report

thank you to giving me the chance to review this exhaustive descriptive review on new insights about diagnosis, treatment and prevention of endometriosis. Despite good overall merit, I have some minors to improve readability:
in the Introduction section, please introduce also rare cases of extra-abdominal endometriosis which merit specific considerations in terms of diagnosis and treatment (i.e. doi: 10.1016/j.jmig.2012.03.005)
please discuss about bowel complaints and urinary dysfunctions, which are very frequently related to endometriosis per se and bowel surgery for endometriosis (i.e. doi:10.1016/j.jmig.2014.05.012)
please modify the reference for the sentence lines 310-312, page 7. Severe ureteral endometriosis is frequent in women with large posterior DIE, parametrical infiltration and low BMI and may require radical ureteral procedures. The preference for one of the different surgical techniques is controversial; some Authors recommend ureteral resection in order to reduce persistence of the disease, risk of reintervention and iatrogenic complications;
please discuss about the different surgical routes for the eradication of recto-vaginal endometriosis with vaginal mucosa infiltration, and in particular about the possibility to isolate the lesion through a vaginal approach;
please discuss about the possibility to detect occult lesions using new technologies such as nir-icg during robotic or laparoscopic procedures

Author Response

Referee 3

thank you to giving me the chance to review this exhaustive descriptive review on new insights about diagnosis, treatment and prevention of endometriosis.

Thank you

Despite good overall merit, I have some minors to improve readability:
in the Introduction section, please introduce also rare cases of extra-abdominal endometriosis which merit specific considerations in terms of diagnosis and treatment (i.e. doi: 10.1016/j.jmig.2012.03.005)

We initially limited this manuscript to pelvic endometriosis since the diagnostic and therapeutic consequences of extra-pelvic endometriosis are not clear yet. However, to emphasise that the pathophysiology also applies to extra-pelvic endometriosis  “Extrapelvic endometriosis occurs in the lungs, although occasionally endometriosis can be found in many other places of the body such as the nose (7)”  was added lines 64-65.

please discuss about bowel complaints and urinary dysfunctions, which are very frequently related to endometriosis per se and bowel surgery for endometriosis (i.e. doi:10.1016/j.jmig.2014.05.012)

Thank you for the suggestion. However, as stated in the last paragraph of the introduction, this manuscript aims to review the impact of the new understanding of the pathophysiology of endometriosis on diagnosis and treatment. Although an excellent article, the changes in bladder and bowel function as a consequence of severe endometriosis do not fit in this manuscript.

please modify the reference for the sentence lines 310-312, page 7. Severe ureteral endometriosis is frequent in women with large posterior DIE, parametrical infiltration and low BMI and may require radical ureteral procedures. The preference for one of the different surgical techniques is controversial; some Authors recommend ureteral resection in order to reduce persistence of the disease, risk of reintervention and iatrogenic complications.

Thank you for this comment. Ref 20 was a mistake and was removed. This paragraph was also extensively revised to focus on the consequences of surgery when fibrosis is not considered part of the endometriosis.

please discuss about the different surgical routes for the eradication of recto-vaginal endometriosis with vaginal mucosa infiltration, and in particular about the possibility to isolate the lesion through a vaginal approach; please discuss about the possibility to detect occult lesions using new technologies such as nir-icg during robotic or laparoscopic procedures

The technical aspects of vaginal cuff surgery if fibrosis is not considered part of the endometriosis are described l 327-329 ‘Also for the vaginal cuff, we prefer a wide excision, with whatever technique,  since in our experience almost all recurrences were observed in the vaginal cuff’,  The use of NIR indocyanine green  to evaluate vascularisation, although important for surgery does not seem to fit in this manuscript.